# DETERMINING THE ETHNO-NATIONALITY OF WRITERS USING WRITTEN ENGLISH TEXT

## ABSTRACT

Ethno-nationality is where nations are defined by a shared heritage, for instance it can be a membership of a common language, nationality, religion or an ethnic ancestry. The main goal of this research is to determine a person's country-of-origin using English text written in less controlled environments, employing Machine Learning (ML) and Natural Language Processing (NLP) techniques. The current literature mainly focuses on determining the native language of English writers and a minimal number of researches have been conducted in determining the country-of-origin of English writers.

Further, most experiments in the literature are mainly based on the TOEFL, ICLE datasets which were collected in more controlled environments (i.e., standard exam answers). Hence, most of the writers try to follow some guidelines and patterns of writing. Subsequently, the creativity, freedom of writing and the insights of writers could be hidden. Thus, we believe it hides the real nativism of the writers. Further, those corpora are not freely available as it involves a high cost of licenses. Thus, the main data corpus used for this research was the International Corpus of English (ICE corpus). Up to this point, none of the researchers have utilised the ICE corpus for the purpose of determining the writers' country-of-origin, even though there is a true potential.

For this research, an overall accuracy of 0.7636 for the flat classification (for all ten countries) and accuracy of 0.6224∼1.000 for sub-categories were received. In addition, the best ML model obtained for the flat classification strategy is linear SVM with SGD optimizer trained with word (1,1) uni-gram model.

## 1 INTRODUCTION

Ethno-nationality can be determined as the identity in which jointly defined by ethnicity and nationality (Sapp, 2012). Identifying a writer's country-of-origin is a part of identifying his ethno-nationality (Jerry, 2008). Foremost focus of ethno-nationality in this research is to identify a writer's country-of-origin based on their writings in a non-native or second language. In recent years, identification of ethno-nationality of a writer has gained a growing interest. In author profiling demographic features (such as age, gender, education, native language, country of origin, etc.,) of an author from a written text will be identified, which are commonly needed in forensic linguistics. For instance, Intelligence to build a profile of their suspect, to identify the author of an anonymous email threat. Therefore, this will enable to limit the search space as well (Estival et al., 2007). Moreover, for business applications this can be useful; for example, in marketing where the demographic features as stated above of customers is important to predict behaviors, upgrade the current products and to develop new products.

People from different ethno-nationality make various language errors when learning a language. Identification of writer's country of origin could have an impact on educational applications designed towards non-native speakers of a language. Besides, this can be used as a plug-in to online tutor systems to provide more tailored feedback to the students about their mistakes (Tetreault et al., 2013). This can help researchers to identify specific teaching and learning issues in different ethno-nationalities. This will enable them to develop pedagogical learning materials to address and solve those issues.

In this research we focus on English as the second language. As English has become a 'universal communication' language due to the globalization. It is no longer restricted to the native countries such as England and United States (Eric, 2013). English language is now available in numerous emerging fields, and has become an essential requirement of labour market and further it is considered to have a cultural importance (Marko, 2009). We define, A controlled environment as a place or area where rules, regulations and norms are subject to strict enforcement. For instance, in an examination setting, candidates are required to produce answers limiting the scope of the question.

## 2 RELATED WORK

Distribution of English language around the world can be identified in two ways, mainly; based on the geographic distribution and based on the Kachru's theory (Braj B, 2004). In this study when segmenting according to the geographic distribution we have considered clusters namely; 'Asia', 'non-Asia', 'South Asia', 'non-South Asia' and 'North America'. According to the Kachru's 'Concentric Circles', English speakers were segmented in to three main categories, namely; 'inner circle', 'outer circle' and 'expanded circle'.

As stated by his model, the inner circle comprises of the countries where the English is the native language (i.e., UK, USA, Canada, New Zealand, etc.). The outer circle includes countries where there are small communities of native English speakers however English is used as the second language in education and official purposes (i.e., Sri Lanka, India, Singapore, Nigeria, etc. ). The expanding circle contains countries which considered English as foreign language (i.e., China, Indonesia, Japan, Saudi Arabia, etc.).

The work of Koppel et al. (2005) is one of the first ethno-nationality identification work where it involves categorizing users based on their native language employing Support Vector Machines (SVMs) on various stylistic features focused on identification of common. Authors achieved 0.802 accuracy using five chosen languages (i.e., Czech, French, Bulgarian, Russian and Spanish) from International Corpus of Learner English (ICLE)(Granger, 2014). The ICLE dataset contains argumentative essays writings of the university students and less nativism and creativity of the writer involved.

Bykh & Meurers (2012) proposed use of recurring n-grams on three different classes (word based, POS based, Open-Class-POS-based) as features for training SVMs. Out of other SVM implementations LIBLINEAR produced best results. For this study, random data from seven native languages selected from the ICLE corpus. The highest performance was obtained for word-level n-grams with an accuracy of 0.8971.

Gebre et al. (2013) employs linear SVM, logistic regression and perceptron (as baseline) for the native language identification (NLI) and achieved accuracy of 0.814 for eleven languages of 'The Test of English as a Foreign Language' (TOEFL) data set(Blanchard et al., 2013). Features used includes; word n-grams, POS n-grams, character n-grams and spelling errors. TOEFL11 has become standard benchmark in NLI tasks since its introduction for the NLI Shared Task 2013. The main limitation of the TOEFL11 dataset is that it is collected in a more controlled environment (i.e., exam for English).

Cimino & Dell'Orletta (2017) utilizes a novel stacked classifier approach where linear logistic regression based sentence feature classifier is stacked with a SVM based document feature classifier with standard lexical, stylistic and syntactic features. However stacked classification approach has gained a minor gain compared to unstacked. Best results of the NLI shared task 2017 reported for this approach with F1-score of 0.8818 for the TOEFL11 dataset.

Kulmizev et al. (2017) introduced 'Groningen' system for the NLI Shared Task 2017 which out performs employing linear SVM for character 1-9 n-grams with the F1-score of 0.8756. Authors have reported that several experiments done with ensemble approach and other features such POS, word, lemma n-grams, skip-grams; and those failed to match the performance of character 1-9 n-gram system.

Goutte & Léger (2017) explored use of voting ensemble SVM models with character, word and POS n-grams. Authors confirms that ensemble methods provide minor but systematic predictive perfor-

Table 1: Summary of the current literature

| Research | Corpus | Accuracy /F1-Score | #Classes | Approach |
|---|---|---|---|---|
| Koppel et al. (2005) | ICLE | 0.802 | 5 | Support Vector Machines (SVMs) on various stylistic features namely; function words, letter n-grams, and errors and idiosyncrasies |
| Bykh & Meurers (2012) | ICLE | 0.8971 | 7 | SVMs with recurring n-grams of three different classes (word based, POS based, Open-Class-POS-based) as features |
| Gebre et al. (2013) | TOEFL11 | 0.814 | 11 | Linear SVM, logistic regressions and perceptron as the linear classifiers with word n-grams, POS n-grams, character n-grams and spelling errors |
| Cimino & Dell'Orletta (2017) | TOEFL11 | 0.8818 /0.8818 | 11 | Two-stacked sentence- and document-feature based classifier architecture. Output of the sentence-level linear regression model is being used by a document-level SVM. |
| Kulmizev et al. (2017) | TOEFL11 | 0.8755 /0.8756 | 11 | Linear SVM and character 1-9 n-grams |
| Goutte & Léger (2017) | TOEFL11 | 0.8736 /0.8740 | 11 | Voting ensemble SVM models approach with character, word and POS n-grams features |
| Markov et al. (2018) | TOEFL11 and ICLE | 0.4883 (TOEFL11) & 0.6948 (ICLE) | 11 (TOEFL11) & 7 (ICLE) | SVM with one-vs-all (OvA) multiclass approach. Abstract POS n-gram and punctuation marks (PM) features have been used. |
| Malmasi & Dras (2018) | TOEFL11, EFCAM-DAT, ASK, JCLC | 0.871 (TOEFL11) | 11 (TOEFL11) | Supervised multi-class classification approach with feature including character, function word, POS n-grams, dependencies, CFG rules, adaptor grammars and TSG fragments used |

mance gains. Highest F1-score performance was with best-vote approach consisting 10 models is 0.8740 for TOEFL11 dataset.

Markov et al. (2018) has used punctuation-based features with POS n-grams for his experiments and accuracies of 0.4883 (TOEFL11) and 0.6948 (ICLE) reported for best performing settings. SVM with one-vs-all (OvA) multi-class classification approach has been used to conduct these experiments.

Malmasi & Dras (2018) employed a supervised multi-class classification approach and incorporated several corpora, including; TOEFL11, EF Cambridge Open Language Database Corpus (EFCAM-DAT), ASK Corpus (Andrespråkskorpus, Second Language Corpus), and Jinan Chinese Learner Corpus. The features extracted were; word/lemma n-grams, character n-grams, function word n-grams, POS n-grams, dependencies, CFG rules, adaptor Grammars and TSG fragments. Highest accuracy of 0.871 is reported for the TOEFL11 dataset.

According to the literature it is evident that most of the researchers in the computational linguistic community have employed ICLE in the early stages and TOEFL in the latest NLI tasks (refer Table 1) which both were collected in more controlled environments. Finally, due to above controlled environments; being unable to capture real nativism in written texts certainly inflate the performance of ethno-nationality identification. On the other hand, identifying the nation of the English writer opposed to his native language will be equally beneficial for author profiling as well. Thus, we

introduce use of International Corpus of English (ICE) for the ethno-nationality identification. To the best of our knowledge, none of the researches have been conducted to determine the writer's country of origin based on the International Corpus of English (ICE) corpus. Moreover, very limited number of researchers have identified the significant features which helps to distinguish Sri Lankan English writers using a large corpus like ICE. One limitation of these researches raised by the authors Tofighı et al. (2012) is that, since for most of the web-based applications, automatic spell-checker has been applied, idiosyncratic features including misspelling and other anomalies are ignored. This may hide some of the features which will be useful in identifying the real categories. Nevertheless, in current context, usage of spell checkers and grammar checkers can be seen frequently. Thus, this will be a common limitation in similar types of researches.

## 3 RESEARCH METHODOLOGY

Supervised learning approach was chosen as it is suitable in classifying text documents into classes more accurately if the classes are known and the data set is labelled (Slotte, 2018). Hence, according to the literature most promising ML algorithms for the text classification such as; Support Vector Machines with Stochastic Gradient Descent Optimizer (linear SVM with SGD), Multinomial Naïve Bayes (MNB), Decision Tree (DT) and Random Forest (an ensemble approach) have been employed. For this research mostly Scikit-learn(Pedregosa et al., 2012) has been used for pre-processing, feature extraction and classification tasks. Scikit-learn is a Python module for machine learning which provides state-of-the-art implementations of many well-known machine learning algorithms, and maintains an easy-to-use interface(Pedregosa et al., 2012). The workflow of the research is depicted in Figure 1 (Left).

The main assumptions of this research are authors from the same ethno-nationality share the same linguistic features in their writing and will often have an influence on the way they express themselves in writings (Jain et al., 2017).

The research questions addressed in this research are;

*Q1. How can texts produced by English writers in a given ethno-nationality be captured from existing corpora?*

*Q3. Which machine learning techniques can gainfully employ the extracted data to identify country-of-origin of English writers?*

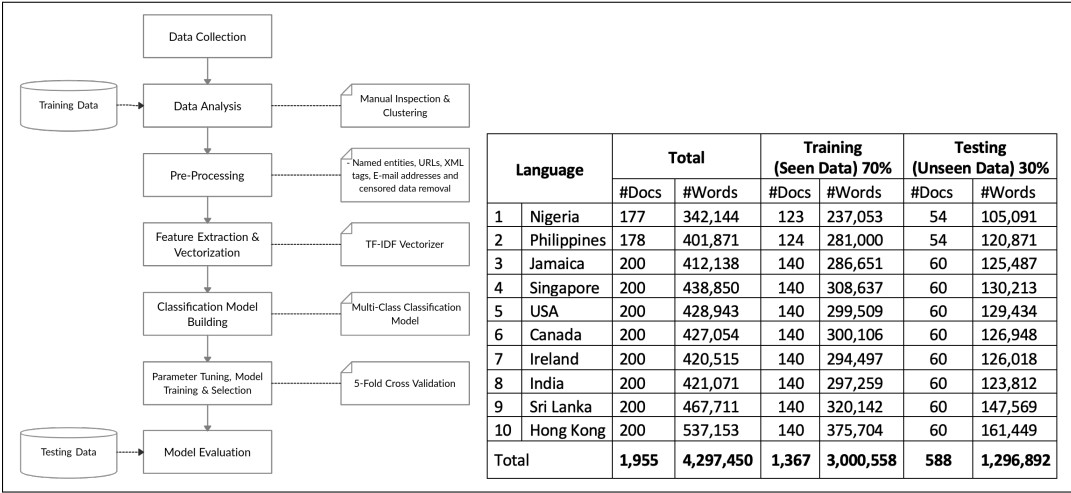

| | | Total | | Training (Seen Data) 70% | | Testing (Unseen Data) 30% | |
|---|---|---|---|---|---|---|---|
| | Language | #Docs | #Words | #Docs | #Words | #Docs | #Words |
| 1 | Nigeria | 177 | 342,144 | 123 | 237,053 | 54 | 105,091 |
| 2 | Philippines | 178 | 401,871 | 124 | 281,000 | 54 | 120,871 |
| 3 | Jamaica | 200 | 412,138 | 140 | 286,651 | 60 | 125,487 |
| 4 | Singapore | 200 | 438,850 | 140 | 308,637 | 60 | 130,213 |
| 5 | USA | 200 | 428,943 | 140 | 299,509 | 60 | 129,434 |
| 6 | Canada | 200 | 427,054 | 140 | 300,106 | 60 | 126,948 |
| 7 | Ireland | 200 | 420,515 | 140 | 294,497 | 60 | 126,018 |
| 8 | India | 200 | 421,071 | 140 | 297,259 | 60 | 123,812 |
| 9 | Sri Lanka | 200 | 467,711 | 140 | 320,142 | 60 | 147,569 |
| 10 | Hong Kong | 200 | 537,153 | 140 | 375,704 | 60 | 161,449 |
| Total | | 1,955 | 4,297,450 | 1,367 | 3,000,558 | 588 | 1,296,892 |

Figure 1: (Left) Workflow of the research methodology. (Right) Document & word distribution of ICE corpus
*#Docs = number of documents, #Words = number of words*

## 3.1 DATA COLLECTION

The main dataset employed in this study is International Corpus of English (ICE) (Greenbaum & Nelson, 1996; Kirk & Nelson, 2018) which comprises of several corpora from different countries. This data-set was built with the intention of providing a resource to conduct comparative studies of English used in different countries where English is the native language or the second language. In order to maintain consistency among each country corpus, common corpus design, corpus size and a common scheme for grammatical annotation have been followed (Kirk & Nelson, 2018). This data-set consists of both written and spoken English texts. Only written texts have been considered for this study. Written texts were gathered from many areas, such as; student writings, letters, academic writing, news reports, instructional writing, persuasive writing and creative writing (Kirk & Nelson, 2018).

The authors and speakers of the texts are aged 18 or above, educated through the medium of English, and were either born in the country in whose corpus they are included or spent the majority of their lives there, or moved there at an early age, and received their education through the medium of English in the country concerned. Written English corpora from Sri Lanka, India, Philippines, Singapore, Canada, Hong Kong, Nigeria, Ireland, Jamaica and USA have been collected and used for this research. Each country is consisted of ≈200 text documents (≈2,000 words per document) (refer Figure 1).

## 3.2 DATA ANALYSIS

As the initial step, 30% (testing data) of the total data of each country corpus was kept as unseen or un-touched data. The remaining 70% of data (training data) was analyzed in order to identify the pre-processing requisites.

## 3.3 DATA PRE-PROCESSING

Raw text-files with specific markups are used as input. These files are cleaned as per the ICE-Corpus markup guide Nelson (2002). In addition, 'strip_accents' parameter for the TF-IDF vectorizer and 'Unicode data normalize NFKD' were used. Moreover, URLs, XML tags, e-mail addresses, censored data, line feeds were removed. The data set contained HTML entity encodings and those were decoded(refer Figure 1). In order to avoid country names, nationality, currency and popular cities being trained as features, all the country specific nationality, cities and country names are removed as a pre-processing step.

## 3.4 CLASSIFICATION MODEL BUILDING

The training data set were used for model building, parameter tuning, training and selection. This labelled training data was tokenized using the TF-IDF vectorizer. Vectorized output was fed to ML classifier to train the model. Further, k-fold (k=3) cross validation technique was used on the training data. Testing data set was kept hold to feed and evaluate the classification model in later stages. Models were built based on two classification strategies;

• **Flat Classification Strategy**

Flat classification refers to a single classifier at the root level as the decision point as depicted in Figure 2(Top). This classifier will handle the all classes as per its classification approaches such as one-vs-rest, one-vs-one...etc. In this study 'one-vs-all' approach has been followed. Hence, the dataset was trained for 10 classes (countries) together.

• **Sub-Category Classification Strategy**

Sub-category classification employs N number of classifiers as depicted in Figure 2(Bottom Left) and Figure 2(Bottom Right). For each sub-category specialized in solving a subset of the problem in which each classifier is trained. Each sub-category was trained separately. As depicted, some are binary (marked with black boxes in Figure 2) and some are multi-class classification models (marked with grey boxes in Figure 2). All the experiments of the sub-categories which were tested are depicted in Figure 3.

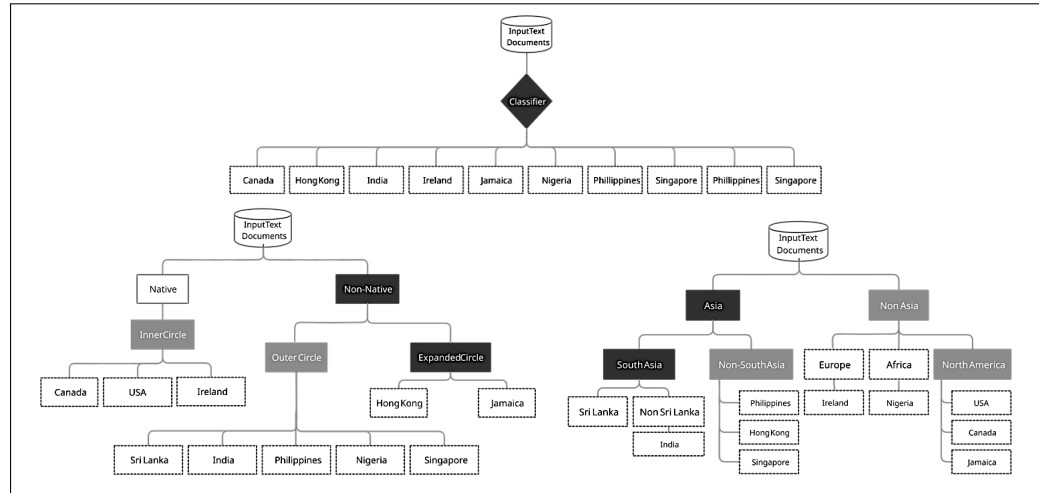

Figure 2: (Top) Flat classification strategy setup. (Bottom Left) Sub-category classification setup based on the Kachru's concentric circles. (Bottom Right) Sub-category classification setup based on the geographical distribution

| DPs/Sub-Category | Type | Description |
|---|---|---|
| **Based on the Kachru's Theory** | | |
| 1 **Native vs Non-Native** | Binary | Native (Inner circle) vs Non-Native (Outer circle + Expanded circle) |
| 2 **Inner Countries** | Multi-class | Inner circle countries (Canada vs USA vs Ireland) |
| 3 **Outer vs Expanded** | Binary | Outer circle vs Expanded circle |
| 4 **Outer Countries** | Multi-class | Outer circle countries (Sri Lanka vs India vs Philippines vs Nigeria vs Singapore) |
| 5 **Expanded Countries** | Binary | Expanded circle countries (Hong Kong vs Jamaica) |
| **Based on the Geographical Distribution** | | |
| 6 **Asia vs Non-Asia** | Binary | Asia class vs Non-Asia class |
| 7 **South Asia vs Non-South Asia** | Binary | South Asia class vs Non-South Asia class |
| 8 **South Asian Countries** | Binary | South Asian countries (Sri Lanka vs India) |
| 9 **Non-South Asian Countries** | Multi-class | Non-South Asian countries (Philippines vs Hong Kong vs Singapore) |
| 10 **Non-Asia** | Multi-class | Non-Asia class (Europe vs Africa vs North America) |
| 11 **North American Countries** | Multi-class | North-American countries (USA vs Canada vs Jamaica) |

Figure 3: Identified sub-categories based on Kachru's Theory and Geographical distribution

## 4 TESTING AND EVALUATION

A combination of both 'hold-out validation' and k-fold cross-validation was used in order to reduce biasness for training and testing data set (Slotte, 2018). Hence, on the 70% of the training data set, 3-fold cross validation technique was used for model building, training and selection and 30% of the data was kept for hold-out validation to validate the machine learning model. For each decision point (where a single classifier needs to be selected) 16 experiments (4 ML x 4 n-grams) have been carried out. For each ML algorithm, word level (1-1,2,3,4) n-grams were tested. Further, performance accuracy, F1-score, precision and recall were calculated to select the best performing ML model.

## 5 RESULTS AND DISCUSSION

**Q1: How can texts produced by English writers in a given ethno-nationality be captured from existing corpora?**

● **Flat classification strategy**

The best ML model obtained for the flat classification strategy is linear SVM with SGD optimizer trained with word (1,1) uni-gram model. Furthermore, overall balanced accuracy of 0.7620 and macro average F1-Score of 0.76 was obtained. Score breakdown for each country is depicted in

Figure 4(Left). In addition, India has the highest F1-score of 0.94 while USA & Sri Lanka holding the lowest F1-score of 0.68. These statistics are further verified by the confusion matrix depicted in Figure 4(Right). Accordingly, all the documents of the India have been classified correctly. The confusion matrix further depicts that the model makes most of the mistakes at the classification of USA vs Jamaica pair. Certainly, one reason could be the geo-graphical proximity between these two countries.

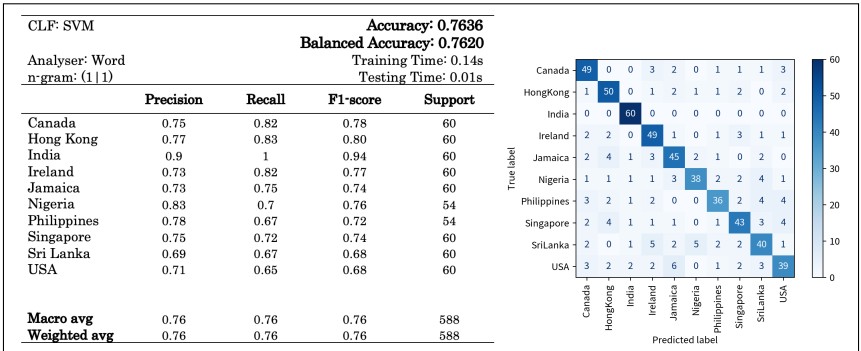

Figure 4: (Left) Test Results for the Flat Classification Strategy. (Right) Confusion matrix for the Flat Classification Strategy

● **Sub-category classification strategy**

In sub-category classification, each sub-category can be perceived as a Decision Point (a single classifier) in the hierarchy of a decision tree. Figure 5 (Left) depicts the test results of the selected ML models for each sub-category. Accuracies of different sub-categories have varied in between 0.6224∼1.000 and F1-score ranges between 0.49∼1.00. Based on the Kachru's concentric theory, lowest F1-Score of 0.71 is for "Outer vs Expanded" sub-category. On the other hand, 'Expanded' sub-category (Hong Kong vs Jamaica) has the highest F1-score of 0.92. In geo-graphical distribution-based model, lowest F1-score of 0.49 is for "Non-Asia" sub-category, while 'South Asian countries' sub-category holding the highest F1-score of 1.0.

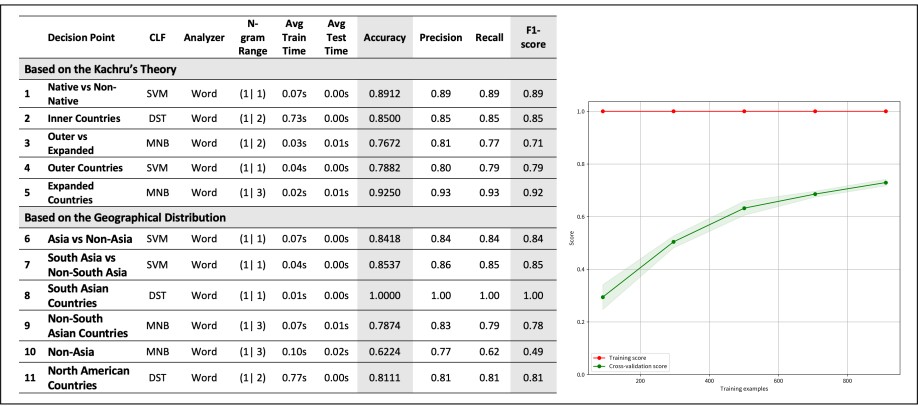

Figure 5: (Left) Test results of the sub-category classification strategy based on selected ML model. (Right) Learning curve for the Flat Classification to assess the generalisability

● **Generalisability of the model**

Assessing the generalizability of the ML models is crucial. Over-fitting leads to poor generalizability of the model. Hence, in order to detect and prevent over-fitting, 'hold-out validation' and k-fold cross-validation used. Further, learning curves on the training data set have been drawn to diagnose whether the model is over-fitting or under-fitting. The red line represents the training score (trained and tested using same data) and the green line represents the cross-validation score(k=3) obtained

for varying number of samples. Cross validated score curve is flattening against the training score curve, as depicted in Figure 5(Right). Thus, this model does not over-fitting or under-fitting for the dataset.

- **Most significant features of the model**

Most significant features identified for the flat classification is depicted in Figure 6. These features are computed base on the coefficients assigned. Some of the features are driven by the cultural and geo-graphical differences of the particular country. For instance, Ireland: northern, christmas, groundwater, queen. and Singapore: business, batik, population. Nigeria: donut(for do not), federal, god..etc.

**y=Canada top features**

| Weight? | Feature |
|---|---|
| +2.129 | canadians |
| +1.966 | employer |
| +1.944 | hannah |
| +1.783 | conventions |
| +1.665 | unclear |
| +1.651 | prairie |
| +1.608 | cegep |
| +1.581 | ash |
| +1.536 | we |
| +1.478 | biotechnology |
| +1.462 | concordia |
| +1.440 | scotia |
| +1.436 | cat |
| +1.411 | sshrc |
| +1.411 | provincial |
| +1.405 | nova |
| +1.362 | ui |
| +1.361 | information |
| +1.349 | groups |
| +1.347 | centre |
| ... 15879 more positive ... | |
| ... 49444 more negative ... | |
| -1.241 | road |
| -1.286 | was |
| -1.405 | mr |
| -1.961 | hope |
| -6.006 | can |

**y=HongKong top features**

| Weight? | Feature |
|---|---|
| +2.151 | government |
| +2.101 | can |
| +2.022 | mr |
| +1.921 | body |
| +1.682 | wong |
| +1.663 | beijing |
| +1.661 | japanese |
| +1.649 | licence |
| +1.639 | mediation |
| +1.611 | yuen |
| +1.574 | political |
| +1.528 | chung |
| +1.509 | ch |
| +1.501 | kowloon |
| +1.468 | hkac |
| +1.446 | capitalism |
| +1.419 | diabetes |
| +1.395 | frank |
| +1.394 | po |
| +1.378 | deviant |
| ... 15358 more positive ... | |
| ... 47655 more negative ... | |
| -1.208 | state |
| -1.557 | this |
| -1.574 | country |
| -1.589 | national |
| -2.037 | that |

**y=India top features**

| Weight? | Feature |
|---|---|
| +1.442 | of |
| +0.817 | mabel |
| +0.806 | singh |
| +0.788 | subhro |
| +0.732 | narasimha |
| +0.722 | college |
| +0.699 | monolina |
| +0.692 | crore |
| +0.679 | duff |
| +0.672 | religion |
| +0.671 | prof |
| +0.670 | annuity |
| +0.659 | books |
| +0.658 | has |
| +0.656 | ayer |
| +0.639 | earth |
| +0.632 | wylie |
| +0.626 | line |
| +0.609 | letter |
| +0.608 | music |
| ... 17822 more positive ... | |
| ... 29508 more negative ... | |
| -0.499 | registrant |
| -0.596 | oil |
| -0.602 | for |
| -0.944 | that |
| -11.279 | in |

**y=Ireland top features**

| Weight? | Feature |
|---|---|
| +3.445 | northern |
| +2.388 | ulster |
| +2.114 | christmas |
| +2.065 | cannabis |
| +1.824 | sign |
| +1.783 | groundwater |
| +1.668 | on |
| +1.628 | ail |
| +1.617 | in |
| +1.563 | narratives |
| +1.520 | garda |
| +1.514 | community |
| +1.503 | kay |
| +1.436 | queen |
| +1.419 | bullet |
| +1.405 | jude |
| +1.386 | xmas |
| +1.346 | ye |
| +1.340 | advice |
| +1.324 | turloughs |
| ... 14395 more positive ... | |
| ... 49248 more negative ... | |
| -1.004 | one |
| -1.009 | program |
| -1.018 | <BIAS> |
| -1.045 | am |
| -1.100 | teachers |

**y=Jamaica top features**

| Weight? | Feature |
|---|---|
| +2.756 | indies |
| +2.386 | st |
| +2.174 | mechtilde |
| +2.113 | am |
| +1.902 | clea |
| +1.803 | hi |
| +1.802 | energy |
| +1.788 | miss |
| +1.764 | seaga |
| +1.760 | west |
| +1.710 | persons |
| +1.632 | linguistics |
| +1.617 | water |
| +1.616 | mi |
| +1.605 | yuh |
| +1.600 | crab |
| +1.569 | campus |
| +1.568 | ml |
| +1.512 | shipping |
| +1.484 | mud |
| ... 14187 more positive ... | |
| ... 49770 more negative ... | |
| -1.141 | state |
| -1.156 | ll |
| -1.306 | lt |
| -1.341 | ve |
| -1.343 | <BIAS> |

**y=Nigeria top features**

| Weight? | Feature |
|---|---|
| +2.381 | state |
| +2.261 | niger |
| +1.891 | corrosion |
| +1.872 | delta |
| +1.770 | kehinde |
| +1.765 | facebook |
| +1.750 | mazi |
| +1.739 | federal |
| +1.651 | donat |
| +1.606 | inec |
| +1.605 | breastfeeding |
| +1.585 | god |
| +1.584 | meaning |
| +1.570 | handicapped |
| +1.547 | drying |
| +1.541 | corruption |
| +1.467 | ekid |
| +1.456 | governor |
| +1.449 | urhobo |
| +1.448 | polio |
| ... 13779 more positive ... | |
| ... 46418 more negative ... | |
| -0.928 | these |
| -0.970 | program |
| -0.981 | don |
| -0.996 | time |
| -1.057 | an |

**y=Philippines top features**

| Weight? | Feature |
|---|---|
| +1.779 | labor |
| +1.732 | cattle |
| +1.701 | la |
| +1.691 | nog |
| +1.654 | romina |
| +1.650 | center |
| +1.627 | edsa |
| +1.610 | metro |
| +1.594 | differance |
| +1.508 | president |
| +1.490 | milf |
| +1.487 | program |
| +1.484 | inez |
| +1.484 | sa |
| +1.459 | mindanao |
| +1.439 | percent |
| +1.422 | ang |
| +1.394 | levinas |
| +1.393 | object |
| +1.393 | influenza |
| ... 15933 more positive ... | |
| ... 51205 more negative ... | |
| -0.961 | programme |
| -1.039 | at |
| -1.084 | real |
| -1.114 | labour |
| -1.307 | students |

**y=Singapore top features**

| Weight? | Feature |
|---|---|
| +1.885 | business |
| +1.807 | batik |
| +1.670 | population |
| +1.646 | editors |
| +1.584 | stuart |
| +1.552 | curriculum |
| +1.550 | tickets |
| +1.525 | jeremy |
| +1.522 | hence |
| +1.504 | lienhwa |
| +1.497 | satan |
| +1.487 | client |
| +1.481 | phonecards |
| +1.476 | tong |
| +1.464 | port |
| +1.458 | postscript |
| +1.448 | national |
| +1.436 | staff |
| +1.394 | straits |
| +1.391 | mental |
| ... 15256 more positive ... | |
| ... 48691 more negative ... | |
| -0.952 | our |
| -1.061 | people |
| -1.069 | summer |
| -1.227 | says |
| -1.853 | we |

**y=SriLanka top features**

| Weight? | Feature |
|---|---|
| +3.198 | ltte |
| +1.965 | plantation |
| +1.956 | in |
| +1.903 | negotiation |
| +1.902 | 2000 |
| +1.896 | fashion |
| +1.808 | tamils |
| +1.796 | unp |
| +1.776 | ngo |
| +1.684 | ceylon |
| +1.658 | baby |
| +1.626 | nanotechnology |
| +1.617 | country |
| +1.598 | internet |
| +1.565 | tea |
| +1.551 | cholesterol |
| +1.525 | mudiyanse |
| +1.512 | india |
| +1.485 | rats |
| +1.449 | with |
| ... 16614 more positive ... | |
| ... 47413 more negative ... | |
| -0.955 | must |
| -0.978 | they |
| -1.106 | for |
| -1.139 | <BIAS> |
| -1.538 | or |

**y=USA top features**

| Weight? | Feature |
|---|---|
| +2.095 | that |
| +1.754 | copyright |
| +1.659 | synderesis |
| +1.646 | program |
| +1.633 | apartment |
| +1.626 | postmodernism |
| +1.508 | stone |
| +1.448 | kids |
| +1.440 | cancer |
| +1.432 | woman |
| +1.355 | photography |
| +1.345 | fp |
| +1.335 | class |
| +1.333 | twilight |
| +1.328 | sommers |
| +1.323 | ireland |
| +1.321 | carbide |
| +1.276 | europe |
| +1.276 | novices |
| +1.251 | lucid |
| ... 15848 more positive ... | |
| ... 44849 more negative ... | |
| -1.411 | was |
| -1.690 | white |
| -2.080 | is |
| -2.134 | university |
| -2.899 | us |

Figure 6: Most significant features of the flat classification

**Q2: Which machine learning techniques can gainfully employ the extracted data to identify country-of-origin of English writers?**

Comparison of ML algorithms based on the selected best parameters are analysed on top of the test data and results are depicted in Figure 7. Linear SVM with SGD optimizer seems to be outperforms in most cases. Moreover, this behavior is also verified in the previous work by Kulmizev et al. (2017), Koppel et al. (2005), Ekaterina (2011), Gebre et al. (2013), Bykh & Meurers (2012). Furthermore, it is noticeable that the DT is under-performing for most cases.

# 6 CONCLUSION AND FUTURE WORK

With various accuracy levels the literature has proven that the ethno-nationality of a person can be identified using their written English texts and this area of research has lot of practical applications and usage. However, as discussed in the literature review still those researches are comprised with lot of limitations as stated above. Therefore, those identified limitations have re-framed this research to obtain solutions to the identified research problem and defined research questions.

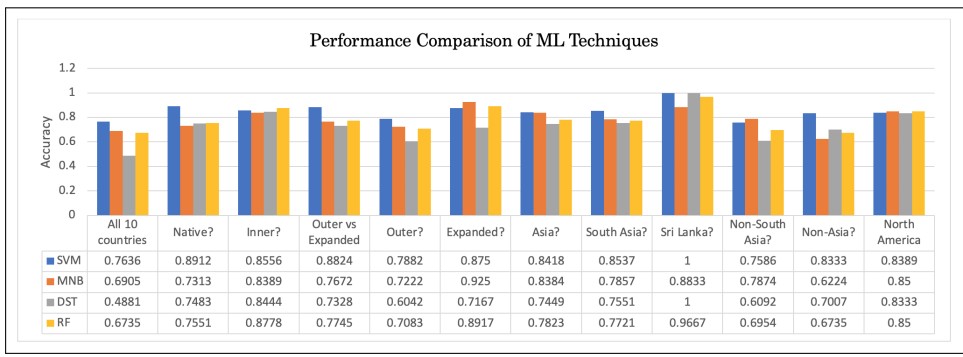

| | All 10 countries | Native? | Inner? | Outer vs Expanded | Outer? | Expanded? | Asia? | South Asia? | Sri Lanka? | Non-South Asia? | Non-Asia? | North America |
|---|---|---|---|---|---|---|---|---|---|---|---|---|
| SVM | 0.7636 | 0.8912 | 0.8556 | 0.8824 | 0.7882 | 0.875 | 0.8418 | 0.8537 | 1 | 0.7586 | 0.8333 | 0.8389 |
| MNB | 0.6905 | 0.7313 | 0.8389 | 0.7672 | 0.7222 | 0.925 | 0.8384 | 0.7857 | 0.8833 | 0.7874 | 0.6224 | 0.85 |
| DST | 0.4881 | 0.7483 | 0.8444 | 0.7328 | 0.6042 | 0.7167 | 0.7449 | 0.7551 | 1 | 0.6092 | 0.7007 | 0.8333 |
| RF | 0.6735 | 0.7551 | 0.8778 | 0.7745 | 0.7083 | 0.8917 | 0.7823 | 0.7721 | 0.9667 | 0.6954 | 0.6735 | 0.85 |

Figure 7: Performance comparison of each ML technique

Our work on Ethno-nationality Identification confirms that linear SVM with SGD optimizer trained with word n-grams can yield a higher level of performance. When determining country-of-origin it is essential to identify set of features which are unique to each author or a particular group of authors.

One spectacular restraint would be the usage of spelling and grammar checkers when writing English. This limitation can be overwritten when using spoken English text. Hence, as a future work transcribed text can be considered to identify the country-of-origin of the English writers. Further, needs to focus more on features which have more pedagogical value and cross corpus generalizability should be assessed to examine the extendibility of the model.

### ACKNOWLEDGMENT

We are deeply grateful for the International English of Corpus (ICE) project owners and maintainers of the respective countries for making available this corpus to the research community without any license fee.

### ETHICS STATEMENT

The Contributors of the respective ICE Corpus countries have informed about the data collection procedures and objectives. All the identifiable named entities of the data have been anonymized in order to disjoint any relation of individuals and organizations to the data. Hence, no one, including the researchers, will be able to link data to a specific individual. Outcomes of these type of researches on categorization of individuals based on ethno-nationality can raise concerns over its usage and discrimination. However, author-profiling is gaining pace and the authors of this paper admire the true potential of such categorizations.

### REPRODUCIBILITY STATEMENT

The implementation details this experiment is available as a supplementary material along with the submission. Kindly note that the supplementary submission does not include the complete dataset. Authors can produce the complete data-set used upon a formal request.

All the raw-data for each country in ICE corpora is available as text files. Files are pre-processed through a jupyter notebook and results were written into a csv called 'ice-merged.csv'. These pre-processing details are also available with the supplementary materials and for further details please refer to README.txt file in the root level.

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
