# OpenReview forum: "Determining the Ethno-nationality of Writers Using Written English Text"
_ICLR.cc/2022/Conference — ICLR 2022 Submitted_

### Official Review · Reviewer_k2RW · 2021-10-29

**Correctness:** 3
**Technical Novelty And Significance:** 1
**Empirical Novelty And Significance:** 1
**Recommendation:** 3
**Confidence:** 5

**Details Of Ethics Concerns:**

The paper is concerned with determining the country of origin of a writer based on the written text. There may be worrisome use cases and applications for such research, although there may also be beneficial use cases. Some of these are discussed in the introduction. The ethics statement briefly refers to these concerns:

"Outcomes of these type of researches on categorization of individuals based on ethno-nationality can raise concerns over its usage and discrimination. However, author-profiling is gaining pace and the authors of this paper admire the true potential of such categorizations."

But this seems like an insufficient discussion of the potential harms.

**Main Review:**

This paper aims to determine the country of origin of English language writers. It uses documents from the International corpus of English (ICE) and employs various learning algorithms and text-based features. The findings are that linear SVM with unigram features works best, that some countries are better distinguished than others, and that some features are more informative than others.

The main premise of this work is that previous work in this space used controlled settings such as English learners texts or standardized examples, while this work examines more naturalistic settings. This is a reasonable motivation. However, besides that, the contributions of this work are very limited: it uses standard algorithms and features, there isn't much analysis of the results, and so the reader does not learn much new from this study. It also doesn't seem like a very good fit for ICLR.

**Summary Of The Paper:**

This paper aims to determine the country of origin of English language writers. It uses documents from the International corpus of English (ICE) and employs various learning algorithms and text-based features. The findings are that linear SVM with unigram features works best, that some countries are better distinguished than others, and that some features are more informative than others.



**Summary Of The Review:**

This paper is an empirical study of standard methods on the problem of classifying a writer's country of origin. It has limited technical contributions or insights.

---

### Official Review · Reviewer_Ak7m · 2021-11-01

**Correctness:** 1
**Technical Novelty And Significance:** 1
**Empirical Novelty And Significance:** 1
**Recommendation:** 3
**Confidence:** 4

**Main Review:**

I think the motivation for studying ethnonationality is a bit weak. The main argument is forensic linguistics and marketing. Both motivations are, in my view, ethically questionable, but the study of ethnonationality also has direct applications for *improving* the fairness of NLP systems.

I like the idea of moving away from TOEFL/ICLE datasets for native language identification - in theory. I previously did research on social media data, and the task suddenly becomes a lot harder. The challenge leaving the TOEFL/ICLE domain is that suddenly a lot of open class words become give-aways of the target. People in Jamaica talk about Jamaica. People in Australia talk about Melbourne, surfing, and kangaroos. In the context of forensic linguists, Australians are not gonna talk about surfing or kangaroos, especially not if they want to hide where they’re from. For this reason, it is common methodology when moving out of a controlled setting (like TOEFL/ICLE), to remove open class words or rely only on character bigram or trigram statistics. The authors fail to do so.

Other problems: a) The protocol is weak; split data temporally rather than randomly to make sure your results will generalize. b) The description of state of the art is out-dated, e.g., see [0-2].

[0] https://aclanthology.org/2020.coling-main.159.pdf
[1] https://aclanthology.org/2020.icon-main.16.pdf
[2] https://arxiv.org/pdf/2010.01869.pdf

**Summary Of The Paper:**

The authors rely on the International Corpus of English to evaluate document classification algorithms on the task of recovering ethnonationality. They evaluate standard algorithms such as decision trees and support vector machines.

**Summary Of The Review:**

Several methodological problems in how data was preprocessed. Motivation is weak, and models are out-dated.

---

### Official Review · Reviewer_bgYR · 2021-11-03

**Correctness:** 2
**Technical Novelty And Significance:** 1
**Empirical Novelty And Significance:** 2
**Recommendation:** 3
**Confidence:** 4

**Main Review:**

Pros:
- The task is of interest

Cons:
- Representing the documents as bags of words is overly simplistic. The experiments and results would normally be considered as baseline.
- There is a line of works by Shuly Wintner that concerns authorship and style modeling in non-native languages, and there may be more related works that are missed here.
- Results must be compared against other approaches, include ablation study etc.

**Summary Of The Paper:**

The work aims to automatically determine the country of origin of authors given their English texts.
Experiments are conducted using the International Corpus of English (ICE corpus), which includes texts of authors from Sri Lanka, India, Philippines, Singapore, Canada, Hong Kong, Nigeria, Ireland, Jamaica and USA; there are 200 text documents per origin country.

It is assumed that authors from the same ethno-nationality share the same `linguistic features'.  The documents are represented as TF-IDF weighted vectors of words, and a classifier is applied (it is not very obvious which classifier exactly, perhaps I missed this detail?).

In classification, the authors apply text classification, either directly into the 10 countries of origin, or using two different class hierarchies.
In addition to multi-class classification, they also conduct binary classification (e.g., native vs. non-native English speakers).

The paper then presents the classification results. Salient features (words) are detailed.


**Summary Of The Review:**

The paper is technically below ICLR threshold.

---

### Official Review · Reviewer_hVHm · 2021-11-04

**Correctness:** 3
**Technical Novelty And Significance:** 1
**Empirical Novelty And Significance:** 1
**Recommendation:** 3
**Confidence:** 4

**Details Of Ethics Concerns:**

We're getting constantly more aware of the concerns associated with profiling people (be it biometrics or some other analytical tools) that can  lead to many harmful applications. The authors do provide an ethical statement, which is in good faith, but I see it as insufficient - limitations and potential harmful usages of this research should have been identified and discussed in more detail.

**Main Review:**

The research problem itself is quite intriguing (but brings many connotations which should have been better explored in the paper), but I really fail to see how this paper fits ICLR's scope and expected submission quality. In other words, there are several key weaknesses to the paper.

1) Starting from the motivation, the authors provide a very vague motivation why detecting ethno-nationality should be more (or equally) important than detecting a native language of the English text writers - how is this problem exactly different than the previous one? Why do we need both, how can both be combined? Do we actually need a different methodology (since the approach is a range of simple classification-based methods) to tackle this problem?
-- The authors claim: "On the other hand, identifying the nation of the English writer opposed to his native language will be equally beneficial for author profiling as well.", but that is such a vague statement - there's no evidence or citations to previous literature underpinning this claim.

2) Suitability to ICLR: I don't see how this paper aligns with ICLR at all. This is in its essence a straightforward application of simple ML-based techniques to a socio-linguistic problem. In essence, this makes it a truly computational linguistics paper, even leaning towards more human-centered linguistic research. It uses some computational tools to study language and linguistic text. There's nothing in the paper advancing representation learning methodology or even advancing our computational tools to study linguistics - such work is tangentially relevant also for conferences such as EMNLP, not to mention more general ML conferences. A good venue for this type of work (not judging its quality, but relevance only) is e.g., COLING.

3) There is no methodological contribution at all: the main contributions of the paper are mostly a redesign of the research problem, and the use of another dataset (which wasn't even collected by the authors, but simply used for a new purpose).

4) The paper puts a lot of emphasis on Related Work, but still bypasses all the more recent work on using Transformer-based neural models for the problem of native language identification from English texts. The coverage of related work stops in 2018, while there are newer papers such as: https://aclanthology.org/2020.coling-main.159.pdf (Lotfi et al., COLING 2020) or another paper: https://aclanthology.org/2020.icon-main.35.pdf

There are more concerns, but these are already sufficient to signal that the paper cannot be accepted in its current form and format.

**Summary Of The Paper:**

This paper presents a simple classification-based approach to a newly defined problem: determining ethno-nationality of writers of English text data. The problem of determining ethno-nationality can be seen as a direct extension of the research problem of identifying the writers' native language based on written English text. The authors define ethno-nationality mostly as country-of-origin and aim to distinguish this demographic feature from native language as another feature. The empirical validations are mostly applications of standard (and even obsolete) classification models on the newly collected data for the introduced problem.

**Summary Of The Review:**

This paper, while tackling an interesting computational linguistic problem, does not bring anything new, and there are detected issues with its thematic fit with ICLR, its lack of novelty and methodological, theoretical or empirical improvements, some problems with presentation (a stronger motivation is needed, the results are presented as snapshots of figures!), etc.

---

### Decision · Program_Chairs · 2022-01-20

**Decision:**

Reject

**Comment:**

This paper proposes to use longstanding statistical learning techniques to identify the nationality of the author of a text.

Reviewers agreed that this work is a poor fit for ICLR, as there is nothing here that advances our understanding of representation learning. Reviewers were further concerned about the soundness of the claims, raising issues about data contamination and comparison with prior work.

Finally, reviewers pointed out (correctly in my view) that work that aims to infer protected identity characteristics of non-user human subjects should be held to an especially high ethical standard, and needs a highly persuasive cost-benefit analysis that defends why the problem is ethical to study at all. The available discussion of ethics is not up to this standard.